# Developing better digital health measures of Parkinson's disease using free living data and a crowdsourced data analysis challenge

**Solveig K. Sieberts**[1]*, **Henryk Borzymowski**[2], **Yuanfang Guan**[3], **Yidi Huang**[4], **Ayala Matzner**[5], **Alex Page**[6,7], **Izhar Bar-Gad**[5], **Brett Beaulieu-Jones**[4,8], **Yuval El-Hanani**[5], **Jann Goschenhofer**[2], **Monica Javidnia**[6,9], **Mark S. Keller**[4], **Yan-chak Li**[10], **Mohammed Saqib**[4], **Greta Smith**[6,9], **Ana Stanescu**[11], **Charles S. Venuto**[6,9], **Robert Zielinski**[6,9], **the BEAT-PD DREAM Challenge Consortium**[¶], **Arun Jayaraman**[12], **Luc J. W. Evers**[13,14], **Luca Foschini**[15], **Alex Mariakakis**[16], **Gaurav Pandey**[10], **Nicholas Shawen**[12,17], **Phil Snyder**[1], **Larsson Omberg**[1]*

1 Sage Bionetworks, Seattle, Washington, United States of America, 2 Independent researcher, 3 Department of Computational Medicine and Bioinformatics, University of Michigan, Ann Arbor, Michigan, United States of America, 4 Department of Biomedical Informatics, Harvard Medical School, Boston, Massachusetts, United States of America, 5 Gonda Brain Research Center, Bar Ilan University, Ramat Gan, Israel, 6 Center for Health + Technology, University of Rochester Medical Center, Rochester, New York, United States of America, 7 Cardiology Division, University of Rochester Medical Center, Rochester, New York, United States of America, 8 Department of Neurology, Brigham and Women's Hospital, Boston, Massachusetts, United States of America, 9 Department of Neurology, University of Rochester, Rochester, New York, United States of America, 10 Department of Genetics and Genomic Sciences, Icahn School of Medicine at Mount Sinai, New York, New York, United States of America, 11 Department of Computing and Mathematics, University of West Georgia, Carrollton, Georgia, United States of America, 12 Center for Rehabilitation Technologies & Outcomes Research, Shirley Ryan AbilityLab, Chicago, Illinois, United States of America, 13 Donders Institute for Brain, Cognition and Behaviour, Department of Neurology, Radboud University Medical Center, Nijmegen, the Netherlands, 14 Institute for Computing and Information Sciences, Radboud University, Nijmegen, the Netherlands, 15 Evidation Health, Santa Barbara, California, United States of America, 16 Department of Computer Science, University of Toronto, Toronto, Ontario, Canada, 17 Medical Scientist Training Program, Northwestern University Feinberg School of Medicine, Chicago, Illinois, United States of America

☯ These authors contributed equally to this work.
¶ Membership of the BEAT-PD DREAM Challenge Consortium is provided in Supporting Information file S1 Acknowledgements.
* solly.sieberts@sagebase.org (SKS); larsson.omberg@sagebionetworks.org (LO)

**Data Availability Statement:** The challenge data are available for researchers who want to improve

## Abstract

One of the promising opportunities of digital health is its potential to lead to more holistic understandings of diseases by interacting with the daily life of patients and through the collection of large amounts of real-world data. Validating and benchmarking indicators of disease severity in the home setting is difficult, however, given the large number of confounders present in the real world and the challenges in collecting ground truth data in the home. Here we leverage two datasets collected from patients with Parkinson's disease, which couples continuous wrist-worn accelerometer data with frequent symptom reports in the home setting, to develop digital biomarkers of symptom severity. Using these data, we performed a public benchmarking challenge in which participants were asked to build measures of severity across 3 symptoms (on/off medication, dyskinesia, and tremor). 42 teams participated and performance was improved over baseline models for each subchallenge.

on models developed during the challenge (https://www.synapse.org/beatpdchallenge).

**Funding:** The BEAT-PD Challenge was funded by the Michael J. Fox Foundation (MJFF) in a grant to LO. MJFF played an advisory role in the design of the challenge. The salaries of SKS, AJ, AM2, NS, PS and LO were partially supported by funds from MJFF. The ensemble computations were partly enabled by Scientific Computing resources at the Icahn School of Medicine at Mount Sinai, and YL and GP were supported by NIH R01HG011407-01A1. YG is funded separately by the Michael J. Fox Foundation. BKBJ was supported by a grant from NIH NINDS (award #: K99NS114850). MSK is supported by a grant from NIH (award #: 5T32HG002295-18). CSV, GS, and RZ received research support funded by the NIH NINDS under award number P50NS108676. These funders had no role in study design, data collection, and analysis, the decision to publish, or preparation of the manuscript.

**Competing interests:** I have read the journal's policy and the authors of this manuscript have the following competing interests: YG serves as scientific advisor for Eli Lilly and Company and Merck & Co.; serves as scientific advisor and receives grants from Merck KGaA. YH has grant funding through BBJ from Sanofi S.A. and UCB for unrelated projects. BBJ has grant funding from Sanofi S.A. and UCB for unrelated projects. AJ is funded by MJ Fox Foundation for data curation. All other authors report no competing interests.

Additional ensemble modeling across submissions further improved performance, and the top models validated in a subset of patients whose symptoms were observed and rated by trained clinicians.

## Author summary

Motion sensors available in consumer devices like smartphones, smartwatches and fitness trackers have enormous potential for use in tracking health and, in the case of movement disorders, understanding symptom severity. In this case, we use data collected from smartphones and smartwatches collected passively as patients go about their daily lives to measure symptom severity in Parkinson's disease. We challenged analysts around the world to develop algorithms to interpret the sensor data from the smart-devices and scored their submissions to determine those that performed the best. 42 teams from around the world participated, and for all 3 symptoms we measured (on/off medication, dyskinesia and tremor) the models using the sensor data showed the ability to predict symptom severity. We also validated these models against symptom severity scores reported by trained doctors.

## Introduction

For many diseases, brief clinic visits do not adequately capture the full lived experience of patients. This is especially true for Parkinson's disease (PD), which is characterized by motor symptoms such as tremors, slowness of movement as well as a broad set of non-motor symptoms in areas such as cognition, mood, and sleep. Of these, only a few are easily evaluated during clinicians exams or captured by patient reports. Because Parkinson's symptoms can be highly variable [1], short, infrequent physician assessments do not capture fluctuations experienced by patients. In fact, motor fluctuations are a common side-effect of the drug treatments commonly used for PD. Additionally, symptoms and assessments that are clinically monitored don't always overlap with the symptoms that interfere with the patient's life [2]. This disconnect is being recognized and, for example, has been noted by the FDA who recently rejected Verily's Virtual Motor Exam for PD as a clinical trial tool because it had "limited capacity to evaluate meaningful aspects of concepts of interest that are relevant to the patients' ability to function in day-to-day life."[3] The development of in-home monitoring using digital health tools, ecological momentary assessments (EMAs) and wearables can offer a way to develop measures of disease that expands the lived experience by collection of real world data [4,5].

Using real world data to better understand the variety and severity of disease requires both exploratory studies as well as validation in a heterogeneous environment. Prior work has demonstrated that digital measures that validate in lab conditions don't always validate in a home environment [6]. Previously, we showed that smartphone sensor measurements from prescribed activities in the home could be used to distinguish PD from non-PD patients [7]. In the same exercise, we also showed that wearable sensors from short, prescribed activities in the clinic could be used to assess symptom severity in PD. We employed a crowd-sourcing approach to achieve these goals and benchmark the best methods [8]. Here we extend our previous work to understand if sensor data, collected passively during patients' daily lives, could be used to assess symptom severity and medication fluctuations. To this end, we ran the Biomarker and Endpoint Assessment to Track Parkinson's Disease (BEAT-PD) DREAM

Challenge which leveraged data coupling patient-reported severity measures from EMAs, with accelerometer data from wrist-worn, consumer smartwatches.

The challenge leveraged two datasets: the Clinicians Input Study (CIS-PD) [9,10] and REAL-PD which is also known as the Parkinson@Home Validation Study [11], both of which employed similar approaches pairing smartwatch sensor data with patient-reported symptom severity collected frequently, at-home, over multiple days. In both studies data from smart watches (Apple Watch in CIS-PD and Motorola Watch with an Android phone REAL-PD) were collected from patients as they went through their daily lives. Patients also reported symptom severity at 30-minute increments using digital Hauser diaries over the course of multiple days of these studies [12]. The challenge leveraged 2,476 symptom reports from 16 subjects for CIS-PD and 782 symptom reports from 12 subjects for REAL-PD.

Challenge participants were asked to build models from wearable data that were able to predict PD severity labels collected through the Hauser diaries. Given the large amount of heterogeneity in disease symptoms between PD patients [13] and large amounts of data available per study subject, we opted to design the challenge such that personalized models of disease could be used to perhaps better capture the previously observed variation in PD [14] and enable a future of patient-specific long term tracking [15].

## Results

### The BEAT-PD DREAM Challenge

We developed three subchallenges using the three symptoms that were captured in both the CIS-PD and REAL-PD Hauser diaries: on/off medication (Subchallenge 1 [SC1]), dyskinesia (Subchallenge 2 [SC2]), and tremor (Subchallenge 3 [SC3]). Challenge participants were free to participate in any or all subchallenges, and one model was scored per team per subchallenge. Challenge participants were asked to predict medication status (SC1) or symptom severity (SC2 & SC3) using non-overlapping 20-minute readings from the sensors associated with the time of the symptom report, as well as baseline patient demographics and MDS-UPDRS scores assessed in both the on and off states by a clinician. Training and test partitions were split within subjects to enable subject-specific models, that is, challenge participants could choose to build either global models or personalized machine learning models (S1 Table & S2 Table). The same training and test splits were used across all three subchallenges. Test partition labels were withheld from challenge participants, and they were asked to predict the phenotype severity in the test partition. Weighted mean-square error (MSE) was used as the scoring metric in each subchallenge and was calculated by a weighted average of the per subject MSE where the weight was the square-root of the subject-specific number of observations in the test set. This weighting scheme was chosen in order to downweight the contributions from subjects with substantially more observations because there was a large range in the number of test observations across subjects (11–99 observations per subject). Models were compared to a baseline *Null* model that generated predictions according to the subject-specific mean of the training labels, which is the best prediction in the absence of any sensor data. Bootstrap p-values were computed to compare each submission to the Null model.

For SC1 (on/off predictions), we received submissions from 37 teams (Fig 1A and S1A Fig), of which 9 submissions performed strictly better than the Null model and 6 performed significantly better at a nominal bootstrap p-value of 0.05. The best model achieved a weighted MSE of 0.878, compared to 0.967 for the Null model. For SC2 (dyskinesia), we received 38 submissions, of which 8 performed strictly better than the Null model and 3 were statistically better at a nominal bootstrap p-value of 0.05 (Fig 1B and S1B Fig). The best model achieved a weighted MSE of 0.405, compared to 0.437 for the Null model. For SC3 (tremor), we received

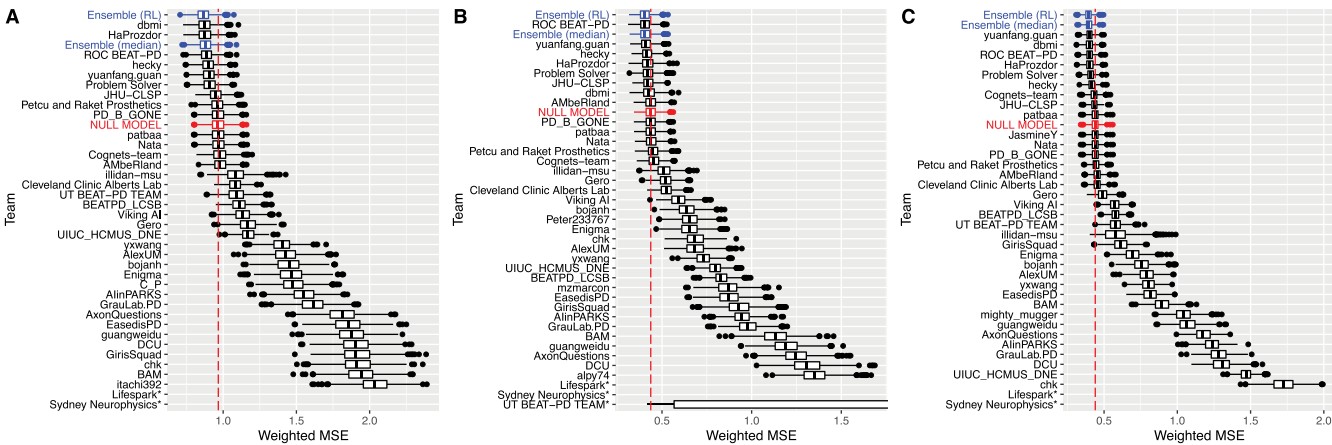

**Fig 1.** Bootstraps (n = 1000) of submissions for (A) SC1: on/off, (B) SC2: dyskinesia, and (C) SC3: tremor. Team models (black) and their ensembles (blue) are ordered by rank. Boxes correspond to the 25th, 50th, and 75th percentiles, and individual points are displayed beyond 1.5*IQR (interquartile range) from the edge of the box. For each sub-challenge, a null model (shown in red) estimated as the subject-specific mean of the training labels was used as a benchmark. Models submitted by teams Lifespark and Sydney Neurophysics were outliers, and have not been displayed in order to present greater definition among the top models. For SC2, the UT BEAT-PD TEAM bootstraps have been truncated for the same reason. The non-truncated figures are available in the Supplementary materials.

submissions from 37 teams, of which 9 strictly outperformed the Null model and 6 were statistically better at nominal bootstrap p-value of 0.05 (Fig 1C and S1C Fig). The two top models were not distinguishable from each other with weighted MSEs of 0.4003 and 0.4006 (p-value of the one-sided Wilcoxon signed-rank test for the bootstrap scores > 0.05), and the weighted MSE of the Null model was 0.440.

Among the top 6 teams whose models performed statistically significantly better than the Null models (with bootstrap p-value < 0.05) in at least one subchallenge, all but one team used signal processing for feature extraction followed by machine learning to build their models. The remaining team, which was the co-winner in SC3 (tremor) and runner-up in SC2 (dyskinesia), input the sensor data directly into a deep learning model. Among the signal processing approaches, two teams fit individual models for each subject: Team dbmi, who won SC1 (on/off) and co-won SC3 (tremor), and Team HaProzdor, who was runner-up in SC1 (on/off). The remaining teams fit global models with subject-specific information to model both within and across subject effects. Along with the lack of consistency among top models, we surveyed all challenge participants and found no association between approaches (including data cleaning and preprocessing, feature extraction, and modeling) and predictive performance.

## Model interpretation

Team dmbi (winner of SC1 and co-winner of SC3) and Team ROC BEAT-PD (winner of SC2) both used random forest-based [16] machine learning modeling which allowed us to explore the model feature importance. Team dbmi trained a random forest model on manually extracted signal features from raw data. Separate models were trained for each patient-phenotype combination. To explore the feature importance within team dbmi's SC1 and SC3 models, we computed SHAP values [17] which quantify the importance of features in a way that is comparable across different models. We computed SHAP values for every prediction and SHAP interaction values for a randomly selected subset of predictions (see Methods). In general, we observed that model predictions were multi-factorial in nature. Effects of individual

features were small, and main effects were generally outweighed by interaction effects (S2 Fig). However, there was general consistency within the top features, even across the two outcomes examined (on/off (SC1) and tremor (SC3)), with the two models sharing 11 of their top 15 features. Nine of the top ten features by SHAP value magnitude correspond to measures of signal magnitude from the accelerometer, including various data quantiles, signal mean, and the 0Hz component of the Fast Fourier Transform (S3 Fig). There was a strong correlation among the top features (S4 Fig), potentially diminishing the importance of individual features and causing stochasticity in feature scores across models. No significant differences in feature effects were observed when comparing across labels (S3 Table). Additionally, we observed no association with previously reported features correlating with Parkinsonian symptoms, such as spectral power in 3-7Hz for tremor or spectral entropy for dyskinesia [18].

For the SC2 (dyskinesia) winning model from team ROC BEAT-PD, model effects were observed to be predominantly linear, so Gini importance [16] was used to explore feature importance in this case. For their models, team ROC BEAT-PD fit a single model for all subjects, incorporating patient characteristics to capture patient heterogeneity. In particular, the clinical MDS-UPDRS scores were highly ranked, suggesting that the predictions were kept grounded by the static/baseline information, then modulated by the real-time sensor data. UPDRS question 4.1 relates to dyskinesia burden, and was by far the most important predictor of dyskinesia level in the CIS-PD cohort, accounting for 45% of the model (S1 Data). A PCA vector based on the UPDRS Part III (motor symptom) questions was the strongest predictor of dyskinesia in the REAL-PD cohort (37% of the model). The most important sensor-based feature in dyskinesia prediction was mean acceleration (vm_mean). Additionally, "counts per minute" (cpm_mean), a feature designed to mimic conventional Actigraph reports, was relatively highly ranked (7% of the dyskinesia model). Finally, correlation coefficients between the acceleration axes (i.e. x vs. y, x vs. z, and y vs. z) were well-represented, with each pair generally accounting for 2–7% of any model. All of these sensor-derived features were more important in the REAL-PD cohort; CIS-PD predictions were largely dictated by the static features. MDS-UPDRS Parts III and IV accounted for over half of the CIS-PD dyskinesia and tremor prediction models.

## Ensemble modeling

To investigate if the overall predictive performance of the challenge could be improved further, we constructed *heterogeneous ensembles* [19] of the solutions to the three subchallenges submitted by the five best-performing teams (ROC, dbmi, HaPrazador, yuanfang.guan and hecky). These teams were selected based on having submitted models that were significantly better than the Null model (nominal bootstrap p-val < 0.05) for at least one subchallenge. One team that met this bar chose not to join this effort and was not included. This investigation was carried out in two stages. In the first stage, several classes of heterogeneous ensemble methods [19–22] were tested in a nested cross-validation setup applied to the training sets of SC1-3 to determine the best ensemble method(s) for each subchallenge. The various ensemble methods showed variable performance across subchallenges when evaluated within the training data cross-validations (S5 Fig), though still outperforming the best individual team model in most cases. This implied that the ensembles were likely to improve the accuracy over the individual models.

Based on their performance in the training data cross-validations, two models were chosen to be evaluated on the test data, one from among the unsupervised methods and one from among the supervised methods. The *median* of the individual base predictions was the best-performing unsupervised ensemble method in the training data evaluation for two of the three

**Table 1. Prediction performance (weighted MSE) of the final supervised and unsupervised ensemble methods on the SC1-3 test sets.** For reference, the performance and name of the winning team in each sub-challenge are also shown.

| Prediction method | SC1 | SC2 | SC3 |
|---|---|---|---|
| Supervised ensemble (RL) | 0.8687 | 0.4048 | 0.3937 |
| Unsupervised ensemble (Median) | 0.8835 | 0.4065 | 0.3978 |
| Winning team's performance | 0.8778 (dbmi) | 0.4053 (ROC BEAT-PD) | 0.4003 (yuanfang.guan) |

subchallenges (S5 Fig). Among supervised ensemble methods, the Reinforcement Learning (*RL*)-based ensemble selection algorithms [20,22] were the best-performing methods for all three subchallenges in the training set evaluation (S5 Fig). In this case, the optimal RL-based algorithm was a $L_2$-regularized linear regression function applied to all five teams' individual predictions. It is important to note that this model selection and optimization was done entirely on the training data set as would have been available to challenge participants, and, in fact, the team generating the ensemble predictions were blinded to the test data in the same way challenge participants were.

We then evaluated the median and RL-based ensemble models in the test data and compared the results to the teams' individual models. The performances of the final ensemble predictors are shown in Table 1 and Fig 1. The RL ensembles were the best performing models in every case, performing better than the median ensembles and best teams' model for every sub-challenge. However, the median ensemble performed better than the best team model in SC3 only. Still, we observe that ensemble approaches can improve prediction accuracy when applied to models submitted during the course of a predictive modeling challenge.

## Subject-level analysis

Using those models statistically outperforming the Null model (from teams dbmi, HaProzdor, hecky, Problem Solver, ROC BEAT-PD, and yuanfang.guan for Subchallenges 1 (on/off) and 3 (tremor), and teams hecky, ROC BEAT-PD, and yuanfang.guan for SC2), we sought to examine whether all subjects were predictable by these models or whether heterogeneity leads to models working well for some patients but not others. To do so, we defined

$$\text{Lift}_{\text{model}} = \text{MSE}_{\text{Null}} - \text{MSE}_{\text{model}}$$

to be the improvement in MSE of the submitted model over the Null model, where a positive value indicates an improved prediction. Generally, we observed that the contribution to MSE improvement over the Null model is largely driven by a small number of subjects, which are well predicted by all or most of the top models (S6 Fig–S8 Fig). For SC1 (on/off), 7 of 22 subjects were responsible for the majority improvement in MSE. Upon examining the percent scale ($\text{Lift}_{\text{model}}/\text{MSE}_{\text{Null}}$), we observed additional subjects who have statistically significant lift, but whose overall contribution to the improvement in performance is low. For SC2 (dyskinesia) and SC3 (tremor), 7 of 16 and 4 of 19 individuals, respectively, account for most of the lift. In rare instances, we observe individuals that are predictable by some models, but are poorly predicted by others (e.g subject 1004 S6 Fig–S8 Fig)). In this case, the teams employing individualized models (dbmi and HaProzdor) perform particularly poorly, suggesting that, in these cases, employing global models protects against overfitting. This observation is consistent across subchallenges.

We evaluated whether subject-specific factors or patient characteristics were associated with better predictability for each model. The patient characteristics explored were age and disease severity as measured by the Movement Disorder Society Unified Parkinson's Disease

Rating Scale (MDS-UPDRS) [23] instrument parts I, II, and IV, as well as part III assessed in both the on- and off-medication state. We also explored the effect of data and metadata characteristics including number of observations (n), the variance of the labels, as well as the mean difference between the symptom reporting period and the time the report was made (reporting lag), with the hypothesis that symptom reports made well after the reporting period may be less accurate. Overall, the only significant association observed was with label variance for on/off medication (S4 Table). The label variance for on/off medication also showed a trend of positive correlation for all models with dyskinesia and tremor (S5 Table and S6 Table).

## Validation of severity as determined by clinician assessment

The top teams were also invited to apply their models to sensor data collected during the completion of short (~30 second) specified tasks for the same study participants in the CIS-PD study. Each of these segments was assessed, in-person, for symptom severity by a clinical PD expert in order to ascertain the degree to which these models recapitulate clinician-rated severity. Four teams (dbmi, HaProzdor, ROC BEAT-PD, and yuanfang.guan) participated in this exercise and submitted predictions for 1277 segments across 16 subjects. Within-subject correlation between the predicted value and the symptom severity label was used as the measure of accuracy, rather than MSE, in order to account for the fact that patients' perception of average severity may differ from a physician's. That is to say, the distributions may be shifted, but we expect the patient- and physician-derived severity ratings to be correlated. For on/off medication, all four models showed significant positive correlation with the clinical ratings for some, but not all, of the subjects (S7 Table–S9 Table). Cross-subject meta-analysis was significant for all teams however. As seen with the challenge predictions, there is a substantial amount of heterogeneity across subjects (4–7 of 14 showing nominal p-value < 0.05) and across models. For a few subjects we see high positive correlations for some teams (dbmi and ROC BEAT-PD) and high negative correlations for others (yuanfang.guan). Interestingly, subject 1004, who showed high heterogeneity across models in the challenge predictions, shows strong positive correlations in the clinical segments.

For tremor (SC3) (S9 Table) and dyskinesia (SC2) (S8 Table), the results were less consistent across models. For tremor, the top scoring model from this subchallenge (yuanfang.gaun) showed only one nominally significant subject (1046, p-value = 0.003), and the cross-subject meta-analysis was not significant after multiple test correction (unadjusted p-value = 0.047). However, the models by dbmi and HaProzdor showed more patients having significant correlation between predictions and labels (6 and 2 of 13, respectively) as well as greater overall significance (meta-analysis p-value = 1.97e-10 and 1.27e-04, respectively). For dyskinesia, only one model (team ROC BEAT-PD) showed nominal significance (unadjusted p-value = 0.035), and only one out of 6 subjects showed significant correlations with any of the models (pearson correlation = 0.286 and 0.290 and unadjusted p-value = 0.003 and 0.004, for ROC BEAT-PD and dbmi, respectively).

## Discussion

The BEAT-PD DREAM Challenge was an open-sourced, collaborative effort to publicly benchmark the ability to use wearable data collected passively during free-living conditions to predict PD symptom severity. Utilizing a challenge framework allowed us to very quickly explore a large space of solutions and engage a community of researchers from around the world to provide solutions. The open source nature of the DREAM challenge frameworks means that all the methods of the participants have been shared and are available as a resource to the community (www.synapse.org/beatpdchallenge). The results of the challenge

demonstrate that passive data from wrist-worn sensors could be used to predict PD symptom severity and motor fluctuations, with multiple models and their ensembles showing significantly improved prediction over the Null model for each symptom tracked. Many of these models showed significant validation against clinical ratings for the same patients. Of the four models which were able to be applied to the short, clinical validation data 4 models in SC1 (on/off), 1 in SC2 (dyskinesia) and 3 in SC3 (tremor) models showed significant association. This is a necessary proof-of-concept toward the development and deployment of validated instruments for passive monitoring of PD. Past efforts have primarily focused on predicting symptom severity from short, well-defined tasks [5,7]. A few efforts have attempted to passively monitor PD symptoms in daily life, chiefly tremor and gait impairments [5,24].

Consistent with previous efforts [7], prediction of dyskinesia was more difficult than prediction of tremor or medication on/off state. This was supported by the fact that only 3 models significantly outperformed the null model for SC2 (dyskinesia), and of those, only one model's predictions significantly correlated with clinician ratings. This worse performance may be due to the difficulty in distinguishing choreic movements from certain types of voluntary movements [18]. Indeed the most important sensor-derived features from the SC2-winning model appear to capture overall motion, rather than specific types of motion. Modeling strategies that take activity into account, for example human activity recognition (HAR) may be more successful in distinguishing movement types, though it is possible that certain types of activities will always be subject to high error rates in the prediction of dyskinesia. This is consistent with previous work that has shown good ability to predict symptom severity in the context of fixed activities [7,18].

Most of the top-performing models used signal processing methods, with the exception of the co-winner of the tremor subchallenge (SC3), which used a deep learning approach. However, it is important to note that a bug discovered in the code of team ROC BEAT-PD would have rendered them the winner in SC3 (tremor) had it been discovered and fixed during the competition (updated weighted MSE = 0.3990). Still, the sole deep learning approach remained among the top models for SC3. It was also the runner-up in SC2, and one of only three models statistically outperforming the null model for prediction of dyskinesia severity (SC2), although it failed to validate in the clinically-rated segments. While deep learning approaches have performed well for predicting PD diagnosis or PD symptom severity in the past, it appears to be most successful when trained on very large data sets, but has performed comparably to signal processing methods in moderate-sized data sets [7]. In this exercise, we also noted a general similarity in prediction across individuals. However, we did observe examples where the deep learning approach performed better or worse than the signal processing approaches (S6 Fig–S8 Fig), although it is presently unclear what factors may drive those differences. In the moderately sized data set used in this Challenge, subject-specific sample size did not appear to be a mediating factor.

Among the signal processing approaches, the top performing approaches utilized a similar workflow—splitting the 20-minute recordings into smaller windows, followed by feature extraction and machine learning; however, there was quite a bit of variability in how these were implemented. Some teams performed some sort of pre-processing (e.g. resampling, normalization, interpolation, removal of gravity, etc) though ROC BEAT-PD (the winner of SC2), did not. Segmentation sizes ranged from 10 to 60 seconds, with varying overlaps. Some used custom features, while several used the publicly available package tsfresh [25]. With respect to machine learning approaches, most teams used random forest [16] models, though one team from among the top performers incorporated these with multiple other models via ensemble approaches to generate their final predictions. There were also differences among the teams in their choice to build individualized versus global models. While both types of models performed similarly overall amongst the top models, there appeared to be examples of patients

where individual models performed substantially better or worse than the global models (S6 Fig–S8 Fig). Given that this is a relatively limited sample size, with respect to the number of individuals, it seems likely that global modeling approaches would benefit greatly from an increase in numbers. Still, given the highly individualistic nature of Parkinson's symptom manifestation, it is unclear whether these types of models will ever be successful in independent individuals without some degree of within-individual training.

We also found that combining information across models in the form of ensemble modeling improved prediction accuracy over the best performing model for all three subchallenges. The RL-based ensemble algorithms [20,22] produced the most accurate predictors for all the subchallenges (Table 1). These near-exhaustive and systematic algorithms are designed to select a parsimonious and effective ensemble (subset) from a large set of base predictors. However, since these algorithms were only applied to five base predictors in this study, the best ensemble was found to be an $L_2$-regularized linear regression function applied to the full set of base predictors. It is also interesting to note that during the course of this analysis, team ROC BEAT-PD discovered a bug in their code, which decreased their weighted MSE to 0.8879 and 0.3990 (from 0.8897 and 0.4012) for SC1 and SC3, though slightly decreasing their MSE in SC2 to 0.4056 from 0.4053. Despite these modest changes, applying the same algorithms to the improved models resulted in little change in the performance of the RL-based ensembles (weighted MSE of 0.8686, 0.4043 and 0.3938 for SC1, SC2 and SC3, respectively, in contrast to 0.8687, 0.4048 and 0.3937 for the RL ensembles of the original submissions) and no change to the median-based ensembles. This demonstrated the robustness of the ensembles. In future Challenges, we aim to apply these ensemble algorithms to larger sets of submissions, and expect to develop even more accurate and parsimonious ensembles.

While the results of this challenge showed promise for the vision of passive- low-burden, at-home monitoring of PD symptoms, the current results are not yet practically useful. Although the symptoms we analyzed are well established motor outcomes, the severity scores available were patient-reported. Patient-reported data can be subject to perception and recall bias, however, researchers have previously observed that patient reporting accuracy is high even in the presence of depression or cognitive difficulties [26]. Additionally, we observed good correspondence between patient and expert severity ratings in the in-clinic (CIS-PD) and at-home clinician visits (REAL-PD). We also found no association between model accuracy and reporting lag in the models developed in the course of this challenge. In our previous challenge we observed that large amounts of data allows for more sophisticated methods [7] to be used. Even though we had large amounts of longitudinal data, it was derived from a small number of subjects (16 to 22 depending on subchallenge). Utilizing a larger number of subjects in future efforts could improve the performance of global models by capturing more of the inter-individual variability expected in PD.

Future studies, such as remote longitudinal studies, have the potential to collect data from thousands of patients [1,27]. If studies like these can be paired with the corresponding outcome variables it might be possible to build better models of disease. Large sample sizes become particularly important as we move away from the basic motor symptoms that are typically measured in the clinic and address additional symptoms that affect patients in their lived experience but are consistently experienced across PD patients.

## Methods

### The CIS-PD Study

The Clinician-Input Study of Parkinson's Disease (CIS-PD) [9,10] was an experiment to assess the utility of Fox Wearable Companion app and accompanying clinician dashboard for

assisting management of patients with PD [9,10]. 51 participants with PD were enrolled across 4 US sites, with 39 patients completing the study. During the 6-month study period, participants wore an Apple Watch Series 2, which continuously collected movement data and streamed it to a cloud server for storage and later analysis. Participants also used the Fox Wearable Companion app to report severity of symptoms and complete digital ON/OFF medication status diaries.

All participants were assessed using the MDS-UPDRS at each in-clinic study visit. Those participating in a substudy completed at the Northwestern University site [28,29] or identified as having significant motor fluctuations, defined in the study as an average of 2 hours a day in an OFF medication state, also completed additional in-clinic, clinician-rated assessments while wearing the smartwatch. These assessments consisted of a series of functional tasks (e.g. drinking water from a cup, folding towels) performed while a trained clinician rated the presence of tremor and dyskinesia for each limb on 0–4 scales. Assessments were done in-person. The criteria for these scales were based on those used in the MDS-UPDRS Part III (Motor) assessment. An assessment of overall severity of motor symptoms was also made for each task on a similar 0–4 scale (0: Normal, 1: Slight, 2: Mild, 3: Moderate, 4: Severe). This series of assessments was performed first while participants were OFF medication (last dose taken the prior calendar day), 30 minutes following the ingestion of their usual dose of medication, and then four additional times at 30 minute intervals. This series of assessments was repeated one additional time at another visit approximately two weeks later, with the patient taking their medication as usual.

All participants completed a paper Hauser diary for the 48 hours prior to the first study visit. Participants with motor fluctuations or participating in the substudy, were also asked to complete electronic symptom diaries at half-hour intervals for 48 hours prior to each of the four study visits. Each diary included self reports, on 0–4 scales, of 3 symptoms: whether the participant felt they were in an ON or OFF medication state, as well as the presence of tremor and dyskinesia. Participants received reminders to complete each diary entry through the Fox Wearable Companion app.

## The REAL-PD Study

The REAL-PD Study, also known as the Parkinson@Home validation study [11], was an experiment designed to assess whether sensor-based analysis of real-life gait can be used to objectively and remotely monitor motor fluctuations in PD. The study recruited 25 neurologist-diagnosed PD patients with motor fluctuations (MDS-UPDRS part IV item 4.3 $\geq$1) and gait impairment (MDS-UPDRS part II item 2.12 $\geq$1 and/or item 2.13 $\geq$1), along with 25 age-matched controls. During home visits, participants were evaluated by a trained assessor, and performed unscripted daily activities while wearing a variety of wearable sensors and while being recorded on video.

Following the in-home visit, PD patients continued to wear a study-provided smartwatch (Motorola Moto 360 Sport with a custom application collecting raw sensor data) on their most affected side and their own Android smartphone in a pant pocket (as available) for two weeks. During this time, they completed various diaries, including a detailed symptom diary at 30-minute intervals over the course of 2 days. The detailed symptom diary asked patients to rate medication status (OFF, ON without dyskinesia, ON with non-troublesome dyskinesia, ON with severe dyskinesia), as well as tremor severity and slowness of gait on a 1–5 scale at each prompt.

## Ethics

CIS-PD was sponsored by the Michael J. Fox Foundation for Parkinson's Research and conducted across four US sites: Northwestern University, the University of Cincinnati, the

University of Rochester, and the University of Alabama at Birmingham. Each site had local Institutional Review Board (IRB)/Research Ethics Board (REB) approval, and all participants signed informed consent.

REAL-PD was sponsored by the Michael J. Fox Foundation for Parkinson's Research. The study protocol was approved by the local medical ethics committee (Commissie Mensgebonden Onderzoek, region Arnhem-Nijmegen, the Netherlands, file number 2016–1776). All participants received verbal and written information about the study protocol and signed a consent form prior to participation, in line with the Declaration of Helsinki.

## The BEAT-PD DREAM Challenge

The Biomarker and Endpoint Assessment to Track Parkinson's Disease (BEAT-PD) DREAM Challenge was launched in January 2020 with the goal of understanding whether passive monitoring with a smartwatch wearable device could be used to monitor PD symptom severity. Challenge participants were provided with a training and a test data set consisting of smartwatch accelerometer data from the REAL-PD and CIS-PD motor fluctuator substudy subjects collected at home during the course of their daily activities. For the subjects in the REAL-PD study, smartwatch gyroscope and smartphone accelerometer data were also provided. Symptom labels for on/off medication (SC1), dyskinesia (SC2) and tremor (SC3) were provided for the training portion of the data, and participants were asked to predict symptom severity in the test portion of the data. Participants could opt to participate in any or all of the subchallenges.

The sensor data were provided as non-overlapping 20-minute segments, each of which corresponded to a patient symptom report. For each segment, the time series data were reported relative to the start of the segment in order to obscure the relative ordering of the segments. The intervening 10-minutes of sensor recordings between each segment were not provided to participants, to prevent reconstruction of segment ordering. Segments showing less than 2 minutes of activity were removed. The training and test data were split randomly for each individual separately, keeping the same within-subject label distributions, in order to facilitate subject-specific modeling. Subjects were filtered on a phenotype-specific basis if they had an insufficient number of observations or label variance. Subjects were included only if they had at least 40 non-missing observations, and at least 2 label categories with 10 or more observations each, or at least 3 label categories with 5 or more observations each. Participants were also provided with minimal demographic information about the patients (age, gender, and race/ethnicity (CIS-PD only)), as well as their MDS-UPDRS scores in the form of totals for Parts I and II, and individual questions for Parts III and IV. For MDS-UPDRS Part III, scores were provided for both the on- and off-medication states.

The training and test sets were split in a 75/25 ratio for each individual separately (S1 Table & S2 Table). For on/off medication, the training set consisted of 1,767 segments from 15 individuals from CIS-PD and 329 segments from 7 individuals from REAL-PD. For dyskinesia, the training set consisted of 1,188 segments from 11 individuals from CIS-PD and 256 segments from 5 individuals from REAL-PD. For tremor, the training set consisted of 1,462 segments for 13 individuals from CIS-PD and 312 segment from 6 individuals from REAL-PD. Participants were also provided with an additional 352 segments for 5 individuals from CIS-PD and 490 segments from 10 individuals from REAL-PD that did not have enough data or variability to be included in the test set, as described above. For testing, the on/off medication data consisted of 587 and 108 segments from CIS-PD and REAL-PD, respectively, the dyskinesia set consisted of 396 and 86 segments from CIS-PD and REAL-PD, respectively, and the tremor set consisted of 487 and 101 segments from CIS-PD and REAL-PD, respectively.

For REAL-PD, most individuals had both smartphone and smartwatch data available for most segments (S10 Table). For CIS-PD, all segments had smartwatch data only.

## Data harmonization

Severity scores were harmonized between the two studies. REAL-PD tremor scores were rescaled to a 0–4 scale from the original 1–5 scale by subtracting one. REAL-PD medication status was converted to binary on/off medication (0/1), and dyskinesia to a 0–3 scale, as shown in S11 Table.

## Submission scoring

Due to the large variation in number of test observations across subjects, we employed a weighted scoring scheme, so that challenge performance wasn't driven primarily by good performance for only a few subjects with large amounts of data. For each subject, $k$, the $MSE_k$ was computed across the test data for that subject. The weighted MSE was then computed as:

$$\text{WMSE} = \frac{\sum_{k=1}^{N} \sqrt{n_k} MSE_k}{\sum_{k=1}^{N} \sqrt{n_k}},$$

where $n_k$ is the number of test observations for individual $k$. Submissions were compared to a baseline "Null" model created using the subject-specific mean of the training labels, which is the best possible prediction in the absence of any sensor data.

Participants were provided the test segments without associated labels and submitted predictions for each. Missing values were not allowed. Submissions were scored using the WMSE described above, and 1,000 bootstraps were performed keeping the same number of within-subject observations for all submissions as well as the Null model. Based on these bootstraps, p-values of each submission versus the Null model were computed as the proportion of iterations in which the Null model outperformed the submission. A nominal (unadjusted) 0.05 p-value was used to select models significantly better than the Null. Additionally, we deemed the top performing model "distinguishable" from the following model using a nominal 0.05 threshold for the p-value of the one-sided Wilcoxon signed-rank test for the bootstrap scores as previously described [30].

## Description of winning methods

**Subchallenges 1 & 3: Team dbmi.** Team dbmi implemented a subject-specific ensemble model using generic time series features. Raw triaxial accelerometer signals were combined using root mean square and partitioned into 30-second windows. They used 552 generic signal features extracted using the tsfresh Python package [25] as model inputs. Windows were assigned the label from the original observation. A random forest was used to model the window labels, and predicted the observation label to be the median of its window predictions. Separate models were tuned and trained for each subject-label combination. They tuned the model hyperparameters using random search over 5-fold cross-validation, and subsequently trained the final contest models using all of the available training data. The same approach was applied to both subchallenges. SHAP values [17] were also computed to estimate the local effects of features on each prediction. They calculated standard SHAP values for each training window, as well as SHAP two-way interaction values for a randomly selected subset of 1000 training windows for each model. To summarize these local effects at a model level, they computed the mean of the magnitude of the SHAP values. They then determined the top 10 features overall by considering the average model-level feature effect over all models.

**Subchallenge 2: Team ROC BEAT-PD.** The 20-minute accelerometer recordings were broken into 30-second windows. 16 features, such as the mean acceleration and dominant frequency, were extracted from each window. Gyroscope data were not used. Simultaneous smartphone and smartwatch sessions were treated as separate recordings, rather than being merged/synchronized. Patient characteristics such as age, gender, and UPDRS scores were also included as features. This way, a general model could be developed, while still allowing personalization for certain (groups of) patients. In total, about 50 features were generated per window. This could be reduced to ~20 using recursive feature elimination [31], but this did not have much impact on model performance, and thus wasn't used. Two random forest regression models were trained: one for CIS-PD and one for REAL-PD. Some overfitting was allowed, so that the models would utilize the sensor data; otherwise, the models tended to lock on to the UPDRS features, yielding relatively static outputs for each participant. The predictions for the 30-second windows were averaged to yield the final prediction for a recording.

**Subchallenge 3: Yuanfang Guan.** This team built a one-dimensional convolutional neural network consisting of five blocks. In each block, there were two convolution layers, each with a filter size of 3, followed by a max pool layer. The last layer used sigmoid activation, and each middle layer used ReLU activation. Cross-entropy was used as the loss function. This model was trained for 100 epochs. 3D rotation augmentation and magnitude augmentation were applied with a factor of [0.8–1.2]. A separate model was built for each individual.

## Analysis of methods used by participants

Challenge participants were sent a survey inquiring about the use of pre-processing and segmentation of the sensor data, feature extraction packages and methods used (if any), machine learning algorithm(s) used, as well as whether individual or global models were used. We received responses from 18 participating teams, all of whom participated in all 3 subchallenges, and tested for associations between the categorical responses and performance using a single-variable linear model for each factor in turn. We found no significant association with any of the factors examined.

## Ensemble model building

To investigate if the overall predictive performance of the challenge could be improved further, we constructed *heterogeneous ensembles* [19] of the solutions to the three challenge problems (SC1-3) submitted by the five best-performing teams (ROC, dbmi, HaPrazador, yuanfang.guan and hecky). Heterogeneous ensembles aggregate diverse base predictive models, such as those developed by the five individual teams for the subchallenges, to build potentially more accurate predictive models [19,21]. To achieve this goal in this challenge, we followed the methodology shown in S9 Fig in order to select the best ensemble models and optimize them, using only the training data. The resulting ensembles were then evaluated on the test set in order to compare them to the models generated by challenge participants. To do so, we implemented a nested cross-validation procedure on each subchallenge separately. In particular, we split the training data into six folds. Five of the folds were used in a traditional cross-validation (CV) framework to generate base predictions from the individual teams and optimize the parameters of each of the ensemble models. The optimized ensemble models were then evaluated on the 6th ("evaluation") fold in order to select the best models.

Using this approach, a variety of ensemble models were built from CV-generated base predictions from the teams' models:

• Simple unsupervised aggregation using the **mean** and **median** of the base predictions.

- **Stacking**: This family of methods learns a meta-predictor over the base predictions. We used 19 standard regression methods from the Python scikit-learn library [32], e.g., LASSO, elastic net, Random Forest, Support Vector Machine and XGBoost, to construct ensembles. In addition to constructing ensembles on the raw predictions, we also trained and evaluated stacking-based ensembles using the following modifications of the raw predictions:

  ○ Normalized values using **Z-scores** to account for variations in scales of the predictions.

  ○ Building separate stacking models for each individual in the training data.

  ○ Combination of the above modifications.

- Caruana et al's ensemble selection (**CES**) algorithm [33,34]: Ensemble selection methods use an iterative strategy to select a parsimonious (small) subset of the base predictors into the final ensemble. Specifically, the CES algorithm considers the complementarity and performance advantage of candidate base predictors to select which one(s) to add to the current ensemble, and terminate the process when there is no improvement in performance.

- Reinforcement Learning (**RL**)-based ensemble selection [20,22]: One of the challenges of the CES algorithm is its greedy, ad-hoc nature, which makes it difficult to generate consistently well-performing ensembles. To address this challenge, members of our team developed several RL [35]-based algorithms [22] that systematically search the space of all possible ensembles to determine the final one. We constructed a variety of supervised ensembles using these RL-based ensemble selection algorithms. Specifically, for each run of this algorithm, we split the training data into 80% to build the RL environment (i.e., the ensemble search space) and used the remaining 20% to learn the RL reward function. Using the information learned during this search process, these algorithms determined the best ensemble.

The parameters of the stacking- and RL-based ensemble algorithms were optimized by executing CV within the first 5 folds using the CVs generated on the participants' models. All the ensemble models were then predicted and evaluated on the evaluation fold. Based on these results, we selected one unsupervised (from among mean and median aggregation) and one supervised (from among the stacking, CES and RL methods) ensemble model to evaluate on the test data. The median aggregation and RL ensembles were the best performing predictors across the three subchallenges. Predictions for these models were then generated on the test set using the full training set, and the performance was evaluated in the test set for comparison against the individual team submissions. The whole ensemble process was implemented in Python.

## Subject-specific analysis

For each model and subject, we computed the Lift measure as follows:

$$\text{Lift}_{\text{model}} = \text{MSE}_{\text{Null}} - \text{MSE}_{\text{model}}.$$

$\text{Lift}_{\text{model}}$ refers to the improvement in MSE of the submitted model over the Null model, where a positive value indicates an improved prediction. 100 bootstrap resamples were used to assess the variance in the Lift, as shown in S6 Fig–S8 Fig. Kendall's tau was used to assess the correlation of lift with the number of test observations (n), reporting lag (mean difference between the reporting time and the time at which the report was made), patient age, MDS-UPDRS parts I, II, and IV, as well as part III assessed in both the on- and -off medication state. Meta-analysis across models was performed by first converting Kendall's tau to Pearson's r [36], and using the metacor function in the "meta" R library [37]. Specifically, the correlations were z-transformed prior to performing a fixed-effect meta-analysis.

## Validation using clinically rated tasks

The in-clinic functional task assessments from the CIS-PD study were used to evaluate whether the models developed by the challenge participants were reflective of clinician-assessed symptom severity. The smartwatch accelerometer data from each subject were segmented into ~30 second segments based on the start and stop time annotations for each activity. This resulted in 1277 segments across 16 subjects, which were provided to the top performing teams in order to generate predictions from their models. The top six teams that statistically outperformed the Null model in at least one subchallenge (dbmi, HaProzdor, hecky, Problem Solver, ROC BEAT-PD, and yuanfang.guan) were invited to participate in this evaluation. One team, Problem Solver, declined to participate, and team hecky was unable to apply their model to such short segments. The remaining teams (dbmi, HaProzdor, ROC BEAT-PD and yuanfang.guan) provided predictions for on/off medication, dyskinesia and tremor severity for the provided segments, which were compared to the clinician ratings. For each subject, the clinician severity ratings for the side (left or right) on which the smartwatch was worn were compared to the participants' predictions. Since the clinicians did not specifically rate on/off medication, overall severity was used as a surrogate against which to compare to the on/off medication predictions. Dyskinesia and tremor predictions were compared against clinician assessments of those symptoms. Accuracy was evaluated within-subject using Pearson's correlation to account for potentially different perceptions of severity between patients, whose ratings were used to train the models, and clinicians. One-sided p-values with $H_A$ = correlation > 0 were reported. For each model, a meta-analysis was performed across subjects using Fisher's log p-value method [38].

## Supporting information

**S1 Table. Number of records (sensor data plus paired label) in the training/test splits for the CIS-PD cohort.**
(PDF)

**S2 Table. Number of records (sensor data plus paired label) in the training/test splits for the REAL-PD cohort.**
(PDF)

**S3 Table. P-value for association of top 10 features from team dbmi with label for on/off, dyskinesia and tremor.**
(PDF)

**S4 Table. Association (Kendall's tau) of subject characteristics with model improvement (on/off medication).**
(PDF)

**S5 Table. Association (Kendall's tau) of subject characteristics with model improvement (dyskinesia).**
(PDF)

**S6 Table. Association (Kendall's tau) of subject characteristics with model improvement (tremor).**
(PDF)

**S7 Table. Validation of models in clinically labeled segments (on/off medication).**
(PDF)

**S8 Table. Validation of models in clinically labeled segments (dyskinesia).**
(PDF)

**S9 Table. Validation of models in clinically labeled segments (tremor).**
(PDF)

**S10 Table. REAL-PD proportion of records with sensor data from the smartphone, the smartwatch or both.**
(PDF)

**S11 Table. REAL-PD medication status harmonization.**
(PDF)

**S1 Fig.** Bootstraps (n = 1000) of submissions for (A) SC1: on/off, (B) SC2: dyskinesia, and (C) SC3: tremor. Team models (black) and their ensembles (blue) are ordered by rank. Boxes correspond to the 25th, 50th, and 75th percentiles, and individual points are displayed beyond 1.5*IQR (interquartile range) from the edge of the box. For each sub-challenge, a null model (shown in red) estimated as the subject-specific mean of the training labels was used as a benchmark.
(PDF)

**S2 Fig. Sum of magnitudes of SHAP interaction values with main and interaction effects separately, shown for the winning models of team dbmi in SC1 and SC3.** Interaction effects outweigh main effects in all models.
(PDF)

**S3 Fig. Distributions of SHAP magnitude for the top 10 features over models for *tremor* and *on_off* labels for the winning models of team dbmi.** *quantile__q_{0.2, 0.3, 0.4, 0.7, 0.8}* are the 20, 30, 40, 70, and 80th data percentiles, respectively; *number_peaks__n_1* is defined as the number of peaks of at least support 1; *sum_values* is the sum over time series values; *mean* is the mean time series value; and *fft_coefficient__coeff_0__attr_{"real", "abs"}* are the real component and absolute value of the 0th coefficient of the fast Fourier Transform (0 Hz), respectively.
(PDF)

**S4 Fig. Heatmap indicating strong correlation among top 10 features from the winning models of team dbmi in SC1 and SC3.** Some features are definitionally equivalent (eg 0 Hz FFT component and mean), while others are very similar in definition (eg 30th and 40th percentiles) contributing to strong correlation.
(PDF)

**S5 Fig. Distributions of performance of the various categories of ensembles constructed for SC1-3 on the corresponding validation (6th) fold of the corresponding training sets.** Also shown are the median scores of each category, as well as those of the five base predictors generated by the individual teams' methods.
(PDF)

**S6 Fig. On/off medication subject-specific lift.** (A) Weighted by sqrt(n) and (B) as a percentage of the null model MSE. Teams are ordered by overall rank.
(PDF)

**S7 Fig. Dyskinesia subject-specific lift.** (A) Weighted by sqrt(n) and (B) as a percentage of the null model MSE. Teams are ordered by overall rank.
(PDF)

**S8 Fig. Tremor subject-specific lift.** (A) Weighted by sqrt(n) and (B) as a percentage of the null model MSE. Teams are ordered by overall rank.
(PDF)

**S9 Fig. Data-driven process used to train and evaluate heterogeneous ensembles for SC1-3.**
(PDF)

**S1 Data. Top 10 features from team ROC BEAT-PD for on/off medication, dyskinesia and tremor.**
(XLSX)

**S1 Acknowledgements. The BEAT-PD DREAM Challenge Consortium.**
(PDF)

## Author Contributions

**Conceptualization:** Solveig K. Sieberts, Arun Jayaraman, Luc J. W. Evers, Luca Foschini, Alex Mariakakis, Nicholas Shawen, Larsson Omberg.

**Data curation:** Solveig K. Sieberts, Phil Synder.

**Formal analysis:** Solveig K. Sieberts, Henryk Borzymowski, Yuanfang Guan, Yidi Huang, Ayala Matzner, Alex Page, Izhar Bar-Gad, Brett Beaulieu-Jones, Yuval El-Hanani, Jann Goschenhofer, Monica Javidnia, Mark S. Keller, Yan-chak Li, Mohammed Saqib, Greta Smith, Ana Stanescu, Charles S. Venuto, Robert Zielinski, Alex Mariakakis, Gaurav Pandey, Nicholas Shawen, Phil Synder.

**Funding acquisition:** Larsson Omberg.

**Investigation:** Solveig K. Sieberts, Arun Jayaraman, Luc J. W. Evers, Gaurav Pandey, Nicholas Shawen.

**Methodology:** Solveig K. Sieberts, Luca Foschini, Gaurav Pandey, Nicholas Shawen.

**Project administration:** Solveig K. Sieberts.

**Software:** Henryk Borzymowski, Phil Synder.

**Supervision:** Solveig K. Sieberts, Larsson Omberg.

**Validation:** Solveig K. Sieberts, Alex Mariakakis, Phil Synder.

**Visualization:** Solveig K. Sieberts.

**Writing – original draft:** Solveig K. Sieberts, Henryk Borzymowski, Yuanfang Guan, Yidi Huang, Ayala Matzner, Alex Page, Izhar Bar-Gad, Brett Beaulieu-Jones, Yuval El-Hanani, Jann Goschenhofer, Monica Javidnia, Mark S. Keller, Yan-chak Li, Mohammed Saqib, Greta Smith, Ana Stanescu, Charles S. Venuto, Robert Zielinski, Luc J. W. Evers, Luca Foschini, Alex Mariakakis, Gaurav Pandey, Nicholas Shawen, Larsson Omberg.

**Writing – review & editing:** Solveig K. Sieberts, Henryk Borzymowski, Yuanfang Guan, Yidi Huang, Ayala Matzner, Alex Page, Izhar Bar-Gad, Brett Beaulieu-Jones, Yuval El-Hanani, Jann Goschenhofer, Monica Javidnia, Mark S. Keller, Yan-chak Li, Mohammed Saqib, Greta Smith, Ana Stanescu, Charles S. Venuto, Robert Zielinski, Luc J. W. Evers, Luca Foschini, Alex Mariakakis, Gaurav Pandey, Nicholas Shawen, Larsson Omberg.

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
