## [Decision Letter · Decision Letter 0]

2 Nov 2022

PDIG-D-22-00272

Developing better digital health measures of Parkinson’s disease using free living data and a crowdsourced data analysis challenge

PLOS Digital Health

Dear Dr. Sieberts,

Thank you for submitting your manuscript to PLOS Digital Health. After careful consideration, we feel that it has merit but does not fully meet PLOS Digital Health's publication criteria as it currently stands. Therefore, we invite you to submit a revised version of the manuscript that addresses the points raised during the review process.

As you can see from the comments below, Reviewer 3 had some additional points to address. Please also improve the clarity of the manuscript in response to Reviewers 1 and 2.

Please submit your revised manuscript within 60 days Jan 01 2023 11:59PM. If you will need more time than this to complete your revisions, please reply to this message or contact the journal office at digitalhealth@plos.org. Please include the following items when submitting your revised manuscript:

We look forward to receiving your revised manuscript.

Kind regards,

Crina Grosan

Academic Editor

PLOS Digital Health

Journal Requirements:

1. Please send a completed 'Competing Interests' statement, including any COIs declared by your co-authors. If you have no competing interests to declare, please state "The authors have declared that no competing interests exist". Otherwise please declare all competing interests beginning with the statement "I have read the journal's policy and the authors of this manuscript have the following competing interests:"

a. State what role the funders took in the study. If the funders had no role in your study, please state: “The funders had no role in study design, data collection and analysis, decision to publish, or preparation of the manuscript.”

b. If any authors received a salary from any of your funders, please state which authors and which funders.

3. We ask that a manuscript source file is provided at Revision. Please upload your manuscript file as a .doc, .docx, .rtf or .tex.

4. Please provide separate figure files in .tif or .eps format only and remove any figures embedded in your manuscript file. Please also ensure that all files are under our size limit of 10MB.

Additional Editor Comments (if provided):

Reviewers' comments:

Reviewer's Responses to Questions

**Comments to the Author**

1. Does this manuscript meet PLOS Digital Health’s publication criteria? Is the manuscript technically sound, and do the data support the conclusions? The manuscript must describe methodologically and ethically rigorous research with conclusions that are appropriately drawn based on the data presented.

Reviewer #1: Yes

Reviewer #2: Yes

Reviewer #3: Partly

2. Has the statistical analysis been performed appropriately and rigorously?

Reviewer #1: Yes

Reviewer #2: Yes

Reviewer #3: I don't know

3. Have the authors made all data underlying the findings in their manuscript fully available (please refer to the Data Availability Statement at the start of the manuscript PDF file)?

Reviewer #1: Yes

Reviewer #2: Yes

Reviewer #3: Yes

4. Is the manuscript presented in an intelligible fashion and written in standard English?

Reviewer #1: Yes

Reviewer #2: Yes

Reviewer #3: Yes

5. Review Comments to the Author

Reviewer #1: This article presents the findings of the Beat-PD challenge. The purpose of this challenge has been to build a model capable of predict symptom severity for Parkinson’s disease based on a wearable sensor (smartwatch). The challenge consisted in three separate tasks, and around 37-38 teams have participated in each task. The article focuses on presenting a summary of the approaches taken by the winning teams, and also proposes an ensemble method consisting in combining the predictions of five-best performing teams.

Overall the article is well-written, but I found difficult to follow the section on model interpretation. Maybe a table summarising the main techniques used by the different teams would help with this. This section seems to give insights into the top models used for task 1 and 2, but not for task 3. 

For the ensemble modelling, the reasoning behind the use of five methods is not well explained : why not 2, or 3, or all the models that were statistically better than the Null model? 

It is also unclear how the train/test partitions were split, if sections were randomly assigned to one of these partitions or the partition was based on some type of temporal aspect: this might have an important impact over the generalisability of the models.

Reviewer #2: This paper presents the results of challenges on various topics, all related to Parkinson disease. The paper is very dense and the results of the challenges are very briefly summarized, in order to preserve an acceptable length of the paper. All code and data are available online, hence readers could eventually reproduce the results of the research, while considering the paper to provide only general guidelines with respect to the actual experiments.

The paper is woth publishing especially by light of its results: in reinforces the idea of mining important information on the health status of a person from affordable common devices (such as smartphone or smartwatch).

I only have a remark. The authors repeatedly use the phrase "statistically better", "significantly better" or "significantly improved", "significantly outperformed" without mentioning the test that was employed.

SHAP values are mentioned and I think it would be useful to introduce them by a short definition, for readers not familiar with the notion.

Reviewer #3: This is an interesting paper attempting to objectively measure Parkinson's disease severity using passive sensor data and making use of a collective effort through a public benchmarking challenge. Enabling objective measures of PD based on passive measurements is important as it does not interfere with activitites of daily living of patients. Few research efforts has been published since it is challenging to make sense of sensor data that could be collected through a routine daily activity (for instance cutting grass with a machine, which could be analyzed by an algorithm as tremor).

The approach taken by the authors is very interesting and at at the same time challenging. Something that I appreciate a lot. 

I have the following comments, which hopefully can guid the authors to improve their work:

- Please explain what is the rationale for including CIS-PD and REAL-PD. How do they complement to each other?

- It is not clear how the Null model has been derived. Please add more details. 

- The work presents five models that performed well. When the results and findings are presented it is not clear to which model do they belong. For instance, Figure S4 presents correlations for top 10 features. For which model/team and which challenge?

- In relation to my previous comment, since all the models/teams results are presented as a reader it is difficult to follow the "line of thought". My suggestion is to decide the best model per challenge and then present results in the following sections.

- When it comes to validation e.g. results presented in tables S8-S10, it is not clear to me why the correlation coefficients for the whole sample are not presented.

- The clinical data was used for validation. It is not clear how were they extracted. For instance, how the clinicians (and how many of them?) observed the video recordings? How did they rate and what did they rate?

- Were the challenges different samples?

- Have the authors considered to assess responsiveness to treatment changes of the model scores? For instance, between OFF, 30 minutes when receiving dose, and follow-up observations.

- Finally, it would be good to add a section on how the authors assessed the validity and reliability of the results they received from the teams?

6. PLOS authors have the option to publish the peer review history of their article (what does this mean?). If published, this will include your full peer review and any attached files.

**Do you want your identity to be public for this peer review?** For information about this choice, including consent withdrawal, please see our Privacy Policy.

Reviewer #1: No

Reviewer #2: No

Reviewer #3: No

---

## [Decision Letter · Decision Letter 1]

7 Feb 2023

Developing better digital health measures of Parkinson’s disease using free living data and a crowdsourced data analysis challenge

PDIG-D-22-00272R1

Dear Dr. Sieberts,

We are pleased to inform you that your manuscript 'Developing better digital health measures of Parkinson’s disease using free living data and a crowdsourced data analysis challenge' has been provisionally accepted for publication in PLOS Digital Health.

Best regards,

Crina Grosan

Academic Editor

PLOS Digital Health

Reviewer Comments (if any, and for reference):

Reviewer's Responses to Questions

**Comments to the Author**

1. If the authors have adequately addressed your comments raised in a previous round of review and you feel that this manuscript is now acceptable for publication, you may indicate that here to bypass the “Comments to the Author” section, enter your conflict of interest statement in the “Confidential to Editor” section, and submit your "Accept" recommendation.

Reviewer #1: All comments have been addressed

Reviewer #2: All comments have been addressed

Reviewer #3: All comments have been addressed

2. Does this manuscript meet PLOS Digital Health’s publication criteria? Is the manuscript technically sound, and do the data support the conclusions? The manuscript must describe methodologically and ethically rigorous research with conclusions that are appropriately drawn based on the data presented.

Reviewer #1: Yes

Reviewer #2: Yes

Reviewer #3: Yes

3. Has the statistical analysis been performed appropriately and rigorously?

Reviewer #1: Yes

Reviewer #2: Yes

Reviewer #3: Yes

4. Have the authors made all data underlying the findings in their manuscript fully available (please refer to the Data Availability Statement at the start of the manuscript PDF file)?

Reviewer #1: (No Response)

Reviewer #2: Yes

Reviewer #3: Yes

5. Is the manuscript presented in an intelligible fashion and written in standard English?

Reviewer #1: Yes

Reviewer #2: Yes

Reviewer #3: Yes

6. Review Comments to the Author

Reviewer #1: (No Response)

Reviewer #2: (No Response)

Reviewer #3: (No Response)

7. PLOS authors have the option to publish the peer review history of their article (what does this mean?). If published, this will include your full peer review and any attached files.

**Do you want your identity to be public for this peer review?** For information about this choice, including consent withdrawal, please see our Privacy Policy.

Reviewer #1: None

Reviewer #2: No

Reviewer #3: No
